# Limb Volume Measurements: A Comparison of Circumferential Techniques and Optoelectronic Systems against Water Displacement

**DOI:** 10.3390/bioengineering11040382

**Published:** 2024-04-15

**Authors:** Giovanni Farina, Manuela Galli, Leonardo Borsari, Andrea Aliverti, Ioannis Th. Paraskevopoulos, Antonella LoMauro

**Affiliations:** 1Istituto Clinico Città Studi di Milano, Via Ampère, 47, 20131 Milan, Italy; giovannifarina.algo@gmail.com; 2Dipartimento di Elettronica, Informazione e Bioingegneria, Politecnico di Milano, Via Giuseppe Ponzio, 34, 20133 Milan, Italy; manuela.galli@polimi.it (M.G.); leonardo.borsari@outlook.com (L.B.); andrea.aliverti@polimi.it (A.A.); 3IGOODI SrL, Via Gaetano Negri, 4, 20123 Milan, Italy

**Keywords:** lymphoedema, limb circumference, optoelectronic plethysmography, 3D bodyscanner, water volumetry, segmental centimetric technique

## Abstract

Background. Accurate measurements of limb volumes are important for clinical reasons. We aimed to assess the reliability and validity of two centimetric and two optoelectronic techniques for limb volume measurements against water volumetry, defined as the gold standard. Methods. Five different measurement methods were executed on the same day for each participant, namely water displacement, fixed-height (circumferences measured every 5 (10) cm for the upper (lower limb) centimetric technique, segmental centimetric technique (circumferences measured according to proportional height), optoelectronic plethysmography (OEP, based on a motion analysis system), and IGOODI Gate body scanner technology (which creates an accurate 3D avatar). Results. A population of 22 (15 lower limbs, 11 upper limbs, 8 unilateral upper limb lymphoedema, and 6 unilateral lower limb lymphoedema) participants was selected. Compared to water displacement, the fixed-height centimetric method, the segmental centimetric method, the OEP, and the IGOODI technique resulted in mean errors of 1.2, 0.86, −16.0, and 0.71%, respectively. The corresponding slopes (and regression coefficients) of the linear regression lines were 1.0002 (0.98), 1.0047 (0.99), 0.874 (0.94) and 0.9966 (0.99). Conclusion. The centimetric methods and the IGOODI system are accurate in measuring limb volume with an error of <2%. It is important to evaluate new objective and reliable techniques to improve diagnostic and follow-up possibilities.

## 1. Introduction

The lymphatic system is a complex of capillaries, lymph vessels and organs within the body that guarantee lymph circulation. Lymph is the fluid that fills the interstices between body cells, and it is composed of an aqueous part and a corpuscular part, mostly represented by lymphocytes.

The main functions of the lymphatic system are: (1) draining the excess liquids and waste substances from the tissues; (2) absorbing the triglycerides (metabolic function) and (3) filtering and blocking the transmission of pathogens (immunologic function) [1,2].

The tissue drainage function prevents the dangerous stagnation of fluid. When this function fails due to injury or dysfunction of the lymphatic system, lymph tends to stagnate and accumulate in the tissues. This condition is called lymphoedema, a chronic swelling of a limb. Lymphoedema can be primary or secondary. Primary or congenital lymphoedema appears early in life and is a clinical manifestation of a congenital defect in the lymphatic vessels or nodes. Secondary lymphoedema is mainly due to damage to the lymphatic system caused by neoplasia, surgery, trauma, infection, or radiotherapy and is a significant cancer survivorship problem [3,4,5,6,7].

Primary lymphedema is rare (1/100.000 individuals), while secondary lymphedema is the most common cause of the disease and affects approximately 1/1000 American people [8,9]. Three hundred million people are estimated to be affected by lymphoedema worldwide. The number of patients is considerable, and the problem cannot be neglected. Epidemiological data from the World Health Organization (WHO) report that 300 million cases of lymphoedema are registered worldwide. Forty-two percent of the cases involve the lower limb. The incidence of secondary lymphoedema, despite the improvements in surgical and radiotherapy techniques, remains high, about 20–30% of patients, either immediately or a few years after surgery; this percentage can reach 60–80% when it is followed by radiation treatments.

Complex Physical Therapy is the only nonsurgical treatment for lymphoedema. The aim of Complex Physical Therapy is firstly to reduce the volume of lymphoedema through the compression bandage and then to maintain such reduction over time [8]. The efficacy of Complex Physical Therapy is usually evaluated by the volumetric measurement of the limbs, which can be direct and indirect [10,11,12,13].

The gold standard is a direct technique called plethysmography or water displacement, which directly measures the limb’s volume after immersion in water. Indirect techniques calculate limb volume by measuring limb circumferences at various levels using a tape. The levels can be identified according to fixed height distances or specific anatomical landmarks (segmental technique). Volume is then calculated using the truncated cone model, to which the various limb segments are assimilated. Volumes are most accurately measured by water displacement [14], although this method is not convenient for routine clinical use, and most operators choose to perform indirect measurements. For practical situations, an easy, objective, and reliable method is still needed.

Accurate volume measurement is important both for evaluating the effectiveness of Complex Physical Therapy as well as classifying the lymphoedema, by comparing the affected limb volume with the contralateral unaffected one.

A very limited number of studies on the validation and comparison of various limb volume measurement methods in patients with lymphoedema are available in the literature to date [15,16,17,18,19,20,21,22]. In the past, bioimpedance spectroscopy, perometry [17,23], infrared optoelectronic volumetry [22], and 3D laser scanners [24,25] were proposed. The intraclass correlation coefficients found were generally strong, ranging from 0.91 to 1 [15,18,22,24,26]. The systematic literature review conducted by Hidding et al. considered 50 studies comparing various methods of limb volume measurement in patients with lymphoedema [17]. Most studies deal with the fixed-height technique, while the segmental technique received little attention. However, the measurement protocols of the studies are very different in terms of fixed-height (5, 8, or 10 cm), geometric approximation of the limb (single truncated cone, succession of truncated cones, or succession of discs) and operator (expert physiotherapist, nurse, or patient). The positioning of the patient (standing, sitting, or lying supine) was seldom considered or even not mentioned, although it may play an important role. All the studies reported by the review considered clinostatism, whether in a sitting or lying down position [17]. There is no literature on orthostatism.

The main objective of this work was to compare different techniques of limb volume measurements with water volumetry as the gold standard, paying particular attention to (1) measurement techniques of limb circumference: fixed-height vs. segmental; (2) positioning: orthostatism vs. clinostatism; and (3) two optoelectronic systems, namely optoelectronic plethysmography based on a motion analysis system and The Gate bodyscanner device that creates 3D avatars with a precise dataset of anthropometric measurements known as Smart Body [27].

## 2. Materials and Methods

The research protocol of this study was approved by the local Research Ethics Committee of Politecnico di Milano (decision no. 07/2023) according to the Declaration of Helsinki. All participants signed a written informed consent form. Exclusion criteria were inability to stand on a single leg, autonomy in moving. and intact limb skin. Each participant underwent all measurements on the same day.

A population of 22 participants (age: 24 (58–70) years; height: 167 (160–179) cm; weight 75 (61–81) Kg) was selected for this prospective study: 7 (age: 24 (23–24) years; height: 181 (167–183) cm; weight 73 (65–80) Kg) were unaffected by lymphoedema, 8 (age: 66 (59–80) years; height: 165 (160–171) cm; weight 75 (61–80) Kg) were affected by unilateral upper limb lymphoedema, 6 (age: 62 (40–75) years; height: 170 (160–179) cm; weight 81 (72–81) Kg) were affected by unilateral lower limb lymphoedema and 1 was affected by unilateral lower and upper limb lymphoedema.

Figure 1 and Figure 2 show all the procedures in orthostatism and clinostatism, respectively.

### 2.1. Centimetric Methods: Fixed-Height and Segmental Techniques

In the fixed-height technique, the upper limb was measured at regular intervals of 5 cm and the lower limb at regular intervals of 10 cm. These heights were marked on both limbs and the limb circumference measurements were recorded at each measurement point in both clinostatism and orthostatism. In the segmental technique, eight anatomical landmarks, coinciding with those used in elastic-compression bandages, were identified for both the upper and lower limbs.

Lower limb. The first point identified was the narrowest point at the supramalleolar level (point B). The second point (point D) was below the styloid apophysis. The third point (E) was measured at the popliteal fossa with the limb bent approximately 30°. The fourth point (point C) is the bulkiest point of the calf muscle identified with the patient in an orthostatic position with the limbs extended. Given the impossibility of objectively identifying the insertion of the gastrocnemius unless assessed by ultrasound, the insertion point of the lower calf muscle (point B1) is identified as an intermediate point between points B and C. Leg length can be calculated as the difference between the heights of points E and B. Once the leg measurement has been completed, the thigh height is measured, starting with the detection of point G, or the gluteal fold. The height of the thigh is then measured as the distance between points E and G. Dividing this height into three equal parts, the intermediate points E1 (1/3 of the thigh height) and F (2/3 of the thigh height) are then identified. Once the eight points were identified on the healthy limb, the heights were replicated on the contralateral pathological limb and the corresponding circumference measurements were taken in both clinostatism and orthostatism (Figure 3).

Upper limb. Measurements of the upper limb were taken by initially placing the patient in an orthostatic position (i.e., seated). The limb was slightly abducted to place the tape inside the axillary cavity to identify point G. The patient was then placed lying supine and point C (or “wrist”) was identified on the flexor surface of the forearm between the hypothenar and thenar folds. Point E (or “elbow”) was identified on the flexural surface at the cubital fold by flexing the forearm on the arm by approximately 90°. The length of the forearm is defined as the distance between points C and E with the limb extended. This length was then divided into four equal parts to identify the following points: point C1 as 1/4 of the height of the forearm from point C, point D as 3/4 of the height of the forearm from point C, and point C2 or the intermediate point between points C1 and D, placed at 1/2 the height of the forearm. The length of the arm was measured as the distance between points E and G. The length of the arm was then divided into three equal parts to identify the following points: point F or the intermediate point of the arm, placed at 2/3 of the height of the arm from point E, and point E1 or the intermediate point between E and F placed at 1/3 of the height of the arm from point E. Once the eight points were identified on the healthy limb, the heights were replicated on the contralateral pathological limb once point C was identified, and the corresponding circumference measurements were taken in both clinostatism and orthostatism (Figure 3).

All these measurements (fixed-height and segmental techniques) were taken by an experienced, senior physiotherapist, with plurennial expertise in lymphoedema evaluation and treatment. After measuring the circumferences, the volume was calculated using the truncated cone model, to which the various limb segments were assimilated, for both methods (fixed-height and segmental techniques) and postures (clinostatism and orthostatism).

### 2.2. Direct Volumetry: Water Displacement Technique

A cylinder of 60 cm × 25 cm × 35 cm filled with 57 L of water was used. The limb was immersed perpendicularly and very slowly in water, to avoid fluid displacements that would invalidate the measurement. Once the immersed limb reached the predetermined point, the water flowing from the spout was collected into a graduated container to quantify the volume of water displaced, equivalent to the volume of the limb segment of interest.

Lower limb. The volume of the foot was firstly measured with the patient sitting and the limb submerged up to the 10 cm mark (for the fixed-height technique) or up to point B (for the segmental technique). Afterwards, the entire limb was immersed in water with the patient standing on the contralateral limb placed on a height-adjustable platform next to the container. Because of the limited size of the container, it was impossible to fully immerse the lower limb, and the first available point above the knee was then considered. This point corresponded to point E1 for the segmental technique or the 50 cm point for the fixed-height technique (Figure 3).

Upper limb. The patient was either in a standing or sitting position (their personal choice based on comfort), with the trunk flexed to allow the immersion of the limb. The volume of the hand was initially determined by immerging it up to the wrist (i.e., point C for the segmental technique or the 0 cm height for the fixed-height technique). Then, the limb was immersed in the container. When full immersion was impossible, the measure was completed until the highest point above the elbow (i.e., E1/F for the segmental technique, see Figure 3).

### 2.3. Optoelectronic Plethysmography: Smart-DX Motion Capture System

Optoelectronic plethysmography uses infrared TV cameras to detect three-dimensional positions of passive reflecting markers attached to the patient according to anatomical points. Once the 3D coordinates of the markers were acquired, a closed-loop graphics surface was defined by connecting the points to form triangles (each marker being one point of the mesh of triangles). The internal volume of the shape is computed using Gauss’ theorem [28]. Because optoelectronic plethysmography is traditionally used for chest wall volumes, dedicated models for limb volume measurements were developed. Markers were placed in correspondence with the anatomical landmarks of the segmental technique. Four models were created, namely (1) upper limb in clinostatism (41 markers), (2) upper limb in orthostatism (49 markers), (3) lower limb in clinostatism (46 markers), and (4) lower limb in orthostatism (61 markers). The number of markers for each of the eight circumferences followed the spatial resolution, being the maximum which allowed two close markers to be recognized as distinct by the system even in the case of a small limb. For both upper and lower limbs, the markers were firstly applied on the patient lying on the bed according to the clinostatic model. The patient was asked to stand up, the markers necessary for the orthostatic model added, and the acquisition in orthostatism was then performed.

### 2.4. IGOODI: The Gate Bodyscanner Technology

IGOODI’s technology, The Gate, is an innovative proprietary scanning cabin. It is a photogrammetry-based 3D reconstruction studio that creates a 3D avatar (i.e., a precise virtual twin of the patient) in an automatic fashion. The process entails a concurrent capture of 128 industrial cameras and sensors that capture height and weight. The scan is completely self-governing, and the user is guided by a virtual avatar assistant through the process. The person, wearing only underwear, is guided to assume the “A-Pose” (i.e., legs slightly apart and arms extended in parallel from the body at a 45° angle, see Figure 1 bottom right panel) as it ensures the best positioning for the 3D reconstruction. The production process of the avatar is composed of various steps: (1) the photogrammetry reconstruction of the 3D model based on the photos, (2) retopology, (3) texturing, (4) rigging, (5) clothing, and (6) the 3D model extraction to graphics engines compatible formats. The measurements were directly taken via IGOODI’s software on the avatar. The marked points were reproduced graphically on the avatar to help the researcher take them in IGOODI’s software at the same points (see Figure 2 bottom right panel). The circumferences of interest were obtained using a virtual tape following the points of the segmental and/or fixed-height techniques, and the volume was computed according to the truncated cone model as for the centimetric methods [27].

### 2.5. Data and Statistical Analysis

All the data were processed using MATLAB (2022. Version R2022a. Natick, MA, USA: The MathWorks Inc.). Data in the text of the Results section and in Table 1 are presented as median (25th percentile–75th percentile). The level of significance was set at *p* < 0.05.

The percent (%) error was calculated as the difference between the analyzed method and the gold standard divided by the gold standard ((method-gold standard/gold standard) × 100). To verify the agreement of two different measurement systems, a correlation analysis was performed between them, and the linear correlation indices (namely, slope (m) and intercept (q) of the interpolation line, indices of accuracy) as well as the coefficient of determination (r^2^, index of dispersion of the data) were estimated. The Bland-Altman graph was also computed.

## 3. Results

### 3.1. Study Population

All participants underwent direct water volumetry and the segmental centimetric method. According to the availability of the person, operator, instrumentation, time and/or cost, 17 participants underwent also the fixed-height centimetric method, 16 participants underwent the optoelectronic system and 9 patients underwent the IGOODI system.

### 3.2. Centimetric Methods: Fixed-Height and Segmental Techniques

The results of the centimetric methods against the gold standard are reported in Figure 4 and Figure 5 and Table 1. Both methods were compared up to the highest point measurable by water volumetry, the point immediately above the elbow or knee (for the fixed-height technique) or point E1/F (for the segmental technique). Although water volumetry can be taken only in orthostatism, it was decided to also compare the clinostatism measurements with it to verify if the position affects the volumes. Both methods provided almost perfect linear regression with the correlation line close to the identity line for both the fixed-height (slope: 1.00023; intercept: −0.01824; r^2^: 0.9865; median difference: 0.038 L; median error: 1.18%) and the segmental (slope: 1.00474; intercept: 0.02162; r^2^: 0.9974; median difference: 0.033 L; median error: 0.86%) techniques. The corresponding parameters computed separately for orthostatism and clinostatism are reported in Table 1.

### 3.3. Optoelectronic Plethysmography

Because the model of the markers followed the same anatomical landmarks identified with the segmental technique, it was decided to compare the measurements with the optoelectronic system with the segmental technique. This allowed the evaluation and comparison of the whole limb. The results are reported in Figure 6, considering both clinostatism and orthostatism. There was a good linear relationship (r^2^ = 0.942), but the optoelectronic system produced underestimated values (slope: 0.874; intercept: −0.1433; median difference: −0.67 L; median error: −16.02%). The Bland-Altman graph indicates that in clinostatism, the error increased with increasing volume of the limb. The corresponding parameters computed separately for orthostatism and clinostatism are reported in Table 1.

### 3.4. IGOODI, The Gate

The results of the IGOODI method against the gold standard are reported in Figure 7 and Table 1. By definition, The Gate provided measurements only in orthostatism with almost perfect linear regression with the correlation line close to the identity line (slope: 0.9966; intercept: 0.054; r^2^: 0.998; median difference: 0.022 L; median error: 0.71%).

## 4. Discussion

In this study, the validity of limb measurement using different circumferential techniques and optoelectronic systems was assessed, with water displacement as the gold standard. Apart from optoelectronic plethysmography, we found percentage errors less than 2%. We have shown that circumferential measurements carried out rigorously, with great carefulness, and by an expert operator can be as accurate as water displacement. However, even under optimal conditions, the centimetric measurements depend on the observer. Instead, we have considered two alternative optoelectronic methods, finding IGOODI to be very accurate, and it appears suitable for the evaluation of limb volume.

Each method considered has advantages and disadvantages (Table 2).

The gold standard was water volumetry [26]. It proved to be difficult to use and affected by possible intrinsic sources of error. Firstly, this technique needs a high level of collaboration from the person being measured during the procedure (i.e., for the immersion and the positioning of the limb), in particular for the lower limb. Patients affected by any exertional limitation of the lower extremity muscles or by any history of impaired ambulation have difficulty immerging the limb perfectly perpendicular to the surface of the water and then staying as still as possible to stabilize the upper level of the immersion. The photo in Figure 1 illustrates this problem and the need for a second operator to help the patient. If the patient were to move, the water would start to oscillate in the container and the measurement could be affected. Secondly, the control of the water temperature is a potential source, as it may induce vasoconstriction or vasodilation of the submerged limb, therefore affecting the total volume [23]. Finally, it is hard to indicate the distribution of the lymphoedema with water volumetry, the measurement time is rather long, and the consumption of water is ecologically unsuitable. For all these reasons, water displacement is not convenient for routine clinical use. Alternative, economic and ecological methods that are comfortable for the patient, easy to perform, and provide an indication of the distribution of the lymphoedema are preferable. We have therefore considered and validated four methods: two centimetric and two optoelectronic.

Circumference measurement is mostly used in clinical practice. We have considered two methods (fixed-height and segmental technique) and two postures (orthostatism and clinostatism), finding errors below 2%. Our errors were lower than those reported in the review conducted by Hidding et al. [17]. The average error of the eight studies using centimetric methods was 6.6%. All the studies deal with the fixed-height technique. Only one study considers the segmental technique, with an average error (1.7%) similar to ours. In line with our results, they also found the segmental technique to perform better than the fixed-height technique [29]. As shown in Table 1, the error found was a bit lower in orthostatism than in clinostatism. We believe that this performance is not due to lower accuracy of the method when performed in clinostatism but to the tissue/muscle relaxation that occurs when the patient is lying on the bed.

Both techniques are economical (as they only require a tape measure as a tool), with results in real time, and they give an idea of the distribution of the lymphoedema. They can be used everywhere (in clinical environments or at the patient’s home).

The centimetric method is an indirect method of volume calculation, as it is based on circumference measurements. The IGOODI system is also an indirect method and proved to be very accurate (error < 1 percent). The measurement is obtained only in orthostatism, and the position is easy to sustain even for old patients (on condition that they keep the standing position without help). The use of avatars was advantageous for the short image acquisition time once the measuring height of circumferences was identified and marked on the skin of the patient. The system did not provide real-time results and the circumferences were calculated posteriorly. The main limitation of this system is that the “The Gate” is available only in one place (at the moment), and therefore it cannot be used in routine clinical practice. The cost of the procedure is higher than for centimetric methods.

Finally, we have also considered a direct method for volume calculation: the optoelectronic system. This system is traditionally used for assessing the respiratory volume changes in the chest wall [30]. We have tried for the first time to extend its use to limb volume. However, the OEP results were not satisfactory. The systematic underestimation of limb volume in both clinostatism (−25.6%) and orthostatism (−8.7%) indicates an important limitation of this method that cannot be introduced in clinical practice. We believe this systematic underestimation is intrinsic to the volume computation of the system. In the geometric models used to compute the volume with optoelectronic plethysmography, straight lines connect the markers. The circumferences are approximated to the polygon inscribed in it. By definition, the inscribed polygon’s area is lower than that of its circumscribed circle. The approximation was even worse in clinostatism (and the measurements were more affected) because fewer markers were used due to the contact areas with the bed. In addition, the measurement procedure is time-consuming due to the application of the markers and data processing, and it requires a laboratory equipped with a motion analysis system. For these reasons, at the moment the optoelectronic plethysmography is not a valid alternative to the other methods evaluated for acquiring static volumes. We first developed the model for the whole limb, but it was not possible to consider the gold standard (because it did not cover the whole limb). We used the segmental technique as a reference. We have not developed the geometrical model for all eight levels, and we did not continue to use the optoelectronic system because of the low quality of the comparison. The first modification that could enhance the accuracy and applicability of optoelectronic plethysmography for limb volume measurements might be the use of more markers in each line. However, increasing the number of markers will reduce the distance between two markers with the risk of approaching the spatial resolution of the system that would not recognize two adjacent markers as separate objects of interest. Another important potential future development can be the use of a laser point to create active (and no longer passive) markers to scan the whole limb. Of note, the use of active markers with optoelectronic plethysmography still needs to be implemented.

However, optoelectronic plethysmography is the only system that can provide a dynamic measurement of the volume variation of the limb. This can be helpful to monitor the blood shift from the trunk to the periphery induced by anesthesia or by respiratory maneuvers (Valsalva or expulsive maneuvers, coughing, etc. [31,32]).

### 4.1. Limitations and Strengths

This study has some limitations. We have considered a single operator for the centimetric evaluations. This was a limitation for the lack of generality of obtained effects. However, we have shown that an expert operator, following a rigorous method of measurement, can perform similarly to the gold standard and/or more complex system (like IGOODI), with an error of ~1%. Future studies should also consider inter-operator reliability to assess the impact of different levels of experience on measurement accuracy. It would be interesting to evaluate the repeatability of the measurement but also the learning process of a non-expert operator following a rigorous method of centimetric measurements.

The number of participants was relatively small, but the dispersion of the data in the correlation plot was very small (with points squeezed around the identity line) and the resulting Pearson correlation coefficient was close to unity, which reinforced the results. The absence of severe lymphoedema was a limitation because increased skin folds, fat deposits, and wart-like growths can develop at this severe stage. The dimorphism becomes so severe (elephantiasis in the worst scenario) that it would be extremely hard to identify the points of interest for all the considered methods. In addition, patients with this kind of severe lymphedema of the lower limb might experience mobility difficulties that would make it challenging to complete the protocol (in particular, the water displacement).

Our study also had several strengths. This was the second study [29] that considered the segmental technique and the first one to include measurements in orthostatism.

Including lymphoedematous limbs and considering both upper and lower limbs was a strength, because we have included a wider range of volumes (ranging from 1 to 7 L) and morphology.

Another important strength was the assessments within the same day, with a maximum of a 1 h delay for the IGOODI method (i.e., the time for the participant to physically reach the system starting from the laboratory where all the other measurements were taken). The volume of the limbs, especially if affected by lymphoedema, could vary considerably even within a few days. The potential variability in limb volume of lymphedema-affected limbs over time is due to many factors: (1) pathophysiology of the condition such as lymphangitis (i.e., inflammation of the lymphatic vessels that causes lymph nodes in the groin or armpit to swell); (2) low-quality treatment (i.e., inappropriate bandage and/or compression garments) so that the swelling associated with the condition is not properly contained; and (3) low compliance of the patient to the treatment who does not wear the prescribed compression garments daily. We exclude temperature as a potential factor of limb volume variability, as arm volumes determined at 38 °C and 16 °C were shown to be almost equal [23]. The comparison among the various methods would then be compromised. Furthermore, it was possible to use the same marks traced on the patient’s limbs during the centimetric technique, whether at fixed heights or segmental, for the position of the passive markers and the virtual measurement on the IGOODI avatar.

### 4.2. Clinical Implications

Measurement of limb volume is helpful for the evaluation and follow-up of oedema not only in patients with lymphedema following cancer treatment [33] but also in those with chronic venous insufficiency.

We have shown that the centimetric techniques, when performed by experienced operators, with the appropriate measuring instruments and following a clear and detailed measurement protocol, are very accurate. This is an important result considering that in 2001, Megens et al. concluded that water displacement volumetry was the only method to provide an accurate estimate of the volume of the upper extremity in women after axillary node dissection for breast cancer [15].

Although the two techniques were competitive in terms of accuracy, we believe that the segmental technique should be preferable for two reasons. First, the segmental data are more comparable within and between patients because the method is based on the identification of anatomical landmarks and proportional standardized distances. The anatomical distribution of lymphoedema, and therefore the identification of dimorphic patterns, can be derived from the segmental technique. Segmental data allowed comparisons between and within participants and different centers. This conclusion was supported by Taylor and colleagues, who concluded that volumes calculated from anatomic landmarks are more accurate than those obtained from circumferential measurements based on distance from fingertips [29].

Second, the proposed segmental technique paralleled the points taken for the elastic-compression garment. This would better quantify and optimize the effect of complex decongestive therapy (i.e., the first intensive phase) and the maintenance phase (i.e., the daily compression by a low-stretch elastic stocking or sleeve) of the treatment by localizing the points that would need more drainage and/or compressive bandaging [10,11,12,13].

The fixed-height technique is good for the follow-up of a single patient, but not in terms of population and dimorphism. The fixed-height technique is single-patient oriented, while the segmental technique is population oriented.

The centimetric measurements are taken in clinostatism during complex decongestive therapy, while the measurements for the elastic compression garment are taken in orthostatism. Evaluating both postures was therefore derived from the clinical procedures.

The potential impact of the findings of this study on patient care and treatment strategies ranges from diagnosis to treatment and follow-up of the patient. Firstly, lymphedema is classified according to the excess percentage volume differences between the affected and unaffected limbs, namely mild (5–20% increase in limb volume), moderate (20–40% increase), or severe (>40% increase) [34]. Secondly, the multi-layered bandage during the first intensive phase of complex decongestive therapy is completely operator-dependent. The multi-layered bandage is repeated daily during complex decongestive therapy. Since the therapist has no tool to measure the pressure developed by the bandage, the only way to verify the efficacy of the treatment is the tracking of the volume variations (hopefully a reduction) of the treated limb. Thirdly, the maintenance phase of the treatment consists of daily compression by a low-stretch elastic stocking or sleeve. Sometimes the compressive garments need to be tailored according to the individual dimorphism. For all these reasons, it is crucially important to provide accurate measurements of the limb size, and therefore to check the mostly used methods and to propose new techniques.

As complex decongestive therapy is still the only available treatment for lymphedema, improving the quality of its two phases through accurate assessment of limb volumes would improve the efficacy of the therapy. An optimized and tailored therapy would ultimately increase the quality of life of patients in terms of aesthetics, increased mobility, and lower infection incidence.

## 5. Conclusions

We have compared four techniques for volume measurements of upper and lower limbs. Firstly, we have considered the most used techniques during complex decongestive therapy (i.e., the fixed-height technique) and maintenance with the compression garment (i.e., the segmental technique). Secondly, we have proposed two innovative optoelectronic technologies, namely optoelectronic plethysmography (based on a motion analysis system) and The Gate body scanner by IGOODI (which provides a 3D avatar of the patient). These methods were compared with water volumetry as the gold standard, finding errors lower than 2%. Although the centimetric method remains the best solution for clinical practice, it is important to propose and evaluate new objective and reliable techniques that can be easily applicable to improve diagnostic and follow-up possibilities.

## Figures and Tables

**Figure 1 bioengineering-11-00382-f001:**
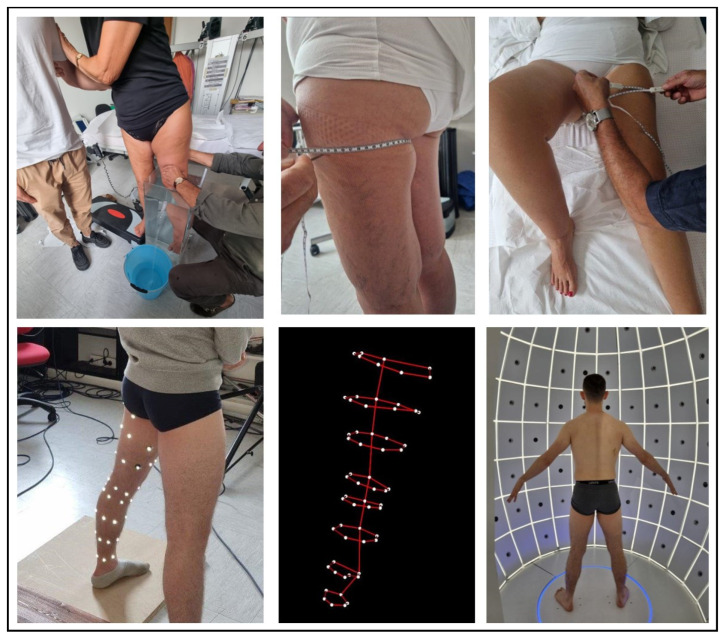
Experimental setup of the different measures of the lower limb: water volumetry (**top left panel**, please notice the patient in need of help to keep the position), circumferential techniques in orthostatism (**top middle panel**) and clinostatism (**top right panel**), optoelectronic plethysmography (**bottom left panel**) and the corresponding 3D marker reconstruction (**bottom middle panel**), and 3D avatar (i.e., a virtual twin of the patient) from the IGOODI Gate technology (**bottom right panel**).

**Figure 2 bioengineering-11-00382-f002:**
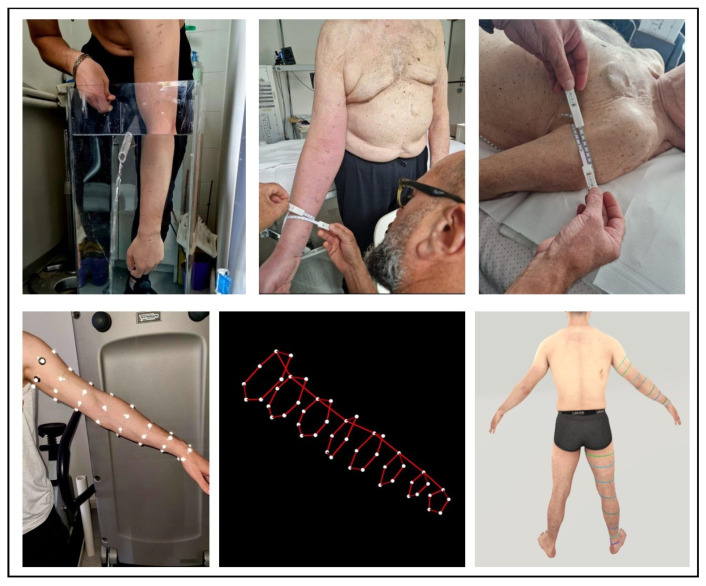
Experimental setup of the different measures of the upper limb: water volumetry (**top left panel**), circumferential techniques in orthostatism (**top middle panel**) and clinostatism (**top right panel**), optoelectronic plethysmography (**bottom left panel**) and the corresponding 3D marker reconstruction (**bottom middle panel**), and 3D avatar (i.e., a virtual twin of the patient) created by the IGOODI Gate technology (**bottom right panel**).

**Figure 3 bioengineering-11-00382-f003:**
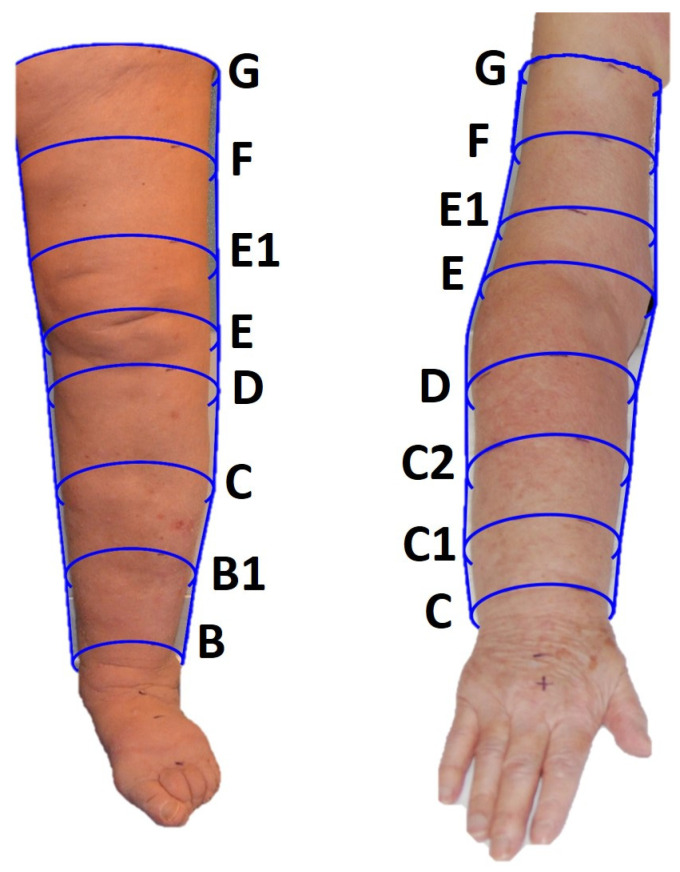
Lower (**left panel**) and upper (**right panel**) limb detection points identified using the segmental technique. The patients signed written informed consent for the publication of their photos.

**Figure 4 bioengineering-11-00382-f004:**
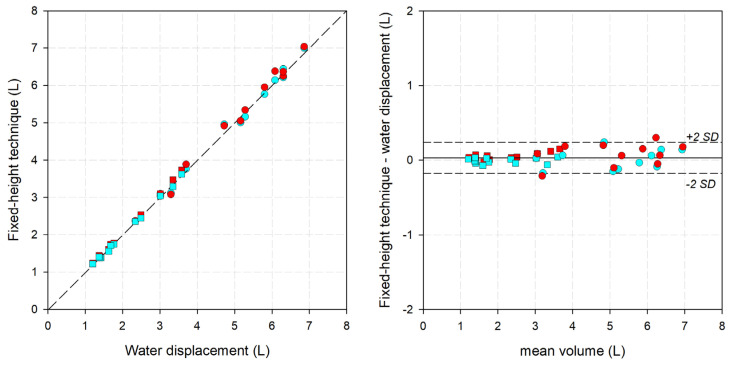
(**Left**) Linear correlation graph, with water displacement on the x-axis and the fixed-height technique on the y-axis. Short dashed line: the interpolating line of the data. (**Right**) Bland-Altman graph, with the average between the two measurements on the x-axis and the difference between the fixed-height technique and water displacement. Solid line: mean difference. Short-dashed lines: mean difference ±2 standard deviations. Circle: lower limb. Square: upper limb. Red: clinostatism. Cyan: orthostatism.

**Figure 5 bioengineering-11-00382-f005:**
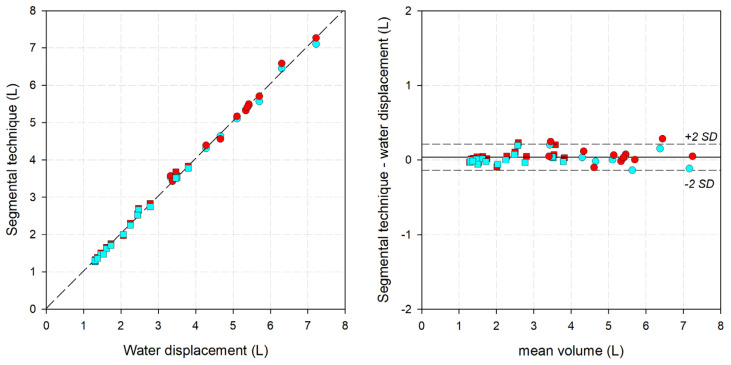
(**Left**) Linear correlation graph, with water displacement on the x-axis and the segmental technique on the y-axis. Short dashed line: the interpolating line of the data. (**Right**) Bland-Altman graph, with the average between the two measurements on the x-axis and the difference between the segmental technique and water displacement. Solid line: mean difference. Short-dashed lines: mean difference ±2 standard deviations. Circle: lower limb. Square: upper limb. Red: clinostatism. Cyan: orthostatism.

**Figure 6 bioengineering-11-00382-f006:**
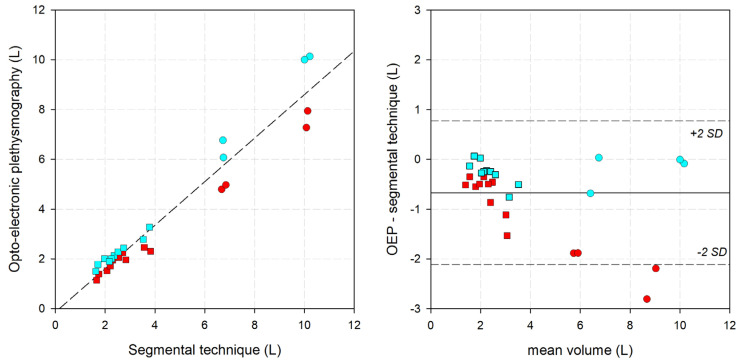
**(Left**) Linear correlation graph, with the segmental technique on the x-axis and the optoelectronic plethysmography data on the y-axis. Short dashed line: the interpolating line of the data. (**Right**) Bland-Altman graph, with the average between the two measurements on the x-axis and the difference between optoelectronic plethysmography and the fixed-height technique. Solid line: mean difference. Short-dashed lines: mean difference ±2 standard deviations. Circle: lower limb. Square: upper limb. Red: clinostatism. Cyan: orthostatism.

**Figure 7 bioengineering-11-00382-f007:**
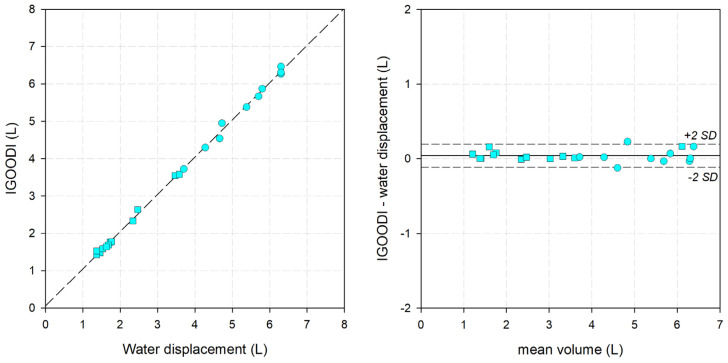
(**Left**) Linear correlation graph, with water displacement on the x-axis and the IGOODI system on the y-axis. Short dashed line: the interpolating line of the data. (**Right**) Bland-Altman graph, with the average between the two measurements on the x-axis and the difference between the IGOODI system and water displacement. Solid line: mean difference. Short-dashed lines: mean difference ±2 standard deviations. Circle: lower limb. Square: upper limb. Cyan: orthostatism.

**Table 1 bioengineering-11-00382-t001:** Mean difference and percentage error (mean, 25th percentile, median, 75th percentile, standard deviation) between each technique and the gold standard and the corresponding parameters of linear correlation.

		Mean Diff. [L]	Mean Error	25° *p*	Median	75° *p*	SD	Slope	Intercept	r^2^	*p*
Fixed-heights technique	clino	0.06	1.79%	0.61%	2.62%	3.62%	2.81%	1.01	0.00	0.997	0.983
	ortho	0.00	−0.13%	−1.74%	0.59%	1.58%	2.42%	1.01	−0.03	0.998	0.983
Segmental technique	clino	0.06	1.78%	0.65%	1.49%	2.80%	3.00%	1.01	0.02	0.997	0.819
	ortho	0.02	0.39%	−1.20%	0.05%	0.96%	2.52%	1.00	0.02	0.998	0.934
Optoelectronic plethysmography system *	clino	−1.11	−25.6%	−30.6%	−26.9%	−20.2%	6.72%	0.74	−0.01	0.987	0.124
	ortho	−0.37	−8.75%	−12.2%	−9.64%	−0.81%	7.52%	0.96	−0.18	0.998	0.132
IGOODI system	ortho	0.02	1.80%	0.06%	0.71%	2.62%	3.04%	0.99	−0.05	0.998	0.824

* the segmental technique set as gold standard.

**Table 2 bioengineering-11-00382-t002:** Advantages and disadvantages of the considered methods.

Method	Advantages	Disadvantages
Water displacement	Gold standard Economical Can be performed anywhere	Long measurement time Onerous for patient and operator The temperature may affect the volume Not-ecological Does not detect lymphoedema distribution
Fixed-heights technique	Very accurate Economical Ecological Can be performed anywhere Detection of lymphoedema distribution Acquisition time shorter than the segmental technique	Poorly reproducible Data not comparable among different patients and clinical centers
Segmental technique	Very accurate Economical Ecological Can be performed anywhere Detection of lymphoedema distribution Highly reproducible Data comparable among different patients and clinical centers	Requires operator experience in identifying anatomical landmarks Long measurement time
Optoelectronic plethysmography system	Highly reproducible Easily comparable data Detection of lymphoedema distribution Short acquisition time	Inaccurate Expensive Needs a laboratory equipped with a motion analysis system Long preparation time Requires experienced operator
IGOODI system	Very accurate Highly reproducible Detection of lymphoedema distribution Short acquisition time	Expensive Use limited to the place where ‘The Gate’ is present

## Data Availability

All data are published in the figures.

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
