# Peer review of "Limb Volume Measurements: A Comparison of Circumferential Techniques and Optoelectronic Systems against Water Displacement"

_bioengineering, 2024, doi:10.3390/bioengineering11040382_

Round 1

Reviewer 1 Report

Comments and Suggestions for Authors

This study evaluated the reliability and validity of four different techniques for measuring limb volume compared to water displacement, considered the gold standard. The techniques included centimetric methods, opto-electronic plethysmography, and a virtual reality-based system. Results showed that centimetric methods and the virtual reality system were accurate, with errors less than 2%. Each method had its advantages and disadvantages, emphasizing the importance of evaluating new techniques for improved diagnostic and follow-up possibilities.

The study is mainly an experimental study. The limb measurements are performed using centimetric methods, opto-electronic plethysmography, and a virtual reality-based system and the results are compared to water displacement and each other. The results indicate the advantages and disadvantages of them.

The experimental results, mainly measurements, are given in tables and graphics. The measurement results are important in the related fields and also for future studies.

The importance of limb measurements using different techniques should be explained better. Also, the studies found in the literature should be discussed better and the contribution of the current study in the literature should be explained better.

Comments on the Quality of English Language

There are some writing and grammar mistakes and they should be corrected and the English writing of the paper should be improved.

Author Response

We thank the reviewer for the time dedicated to our manuscript and for the suggestions that we have tried to implement in the new version.

The importance of limb measurements using different techniques should be explained better.

Also, the studies found in the literature should be discussed better and the contribution of the current study in the literature should be explained better.

We have now discussed this as follows: “The potential impact of the findings of this study on patient care and treatment strategies ranges from the diagnosis to the treatment and the follow-up of the patient. Firstly, lymphedema is classified according to the excess percentage volume differences between the affected and unaffected limbs, namely mild (5-20% increase in limb volume), moderate (20-40% increase), or severe (>40% increase)[33]. Secondly, the multi-layered bandage during the first intensive phase of the complex decongestive therapy is completely opera-tor-dependent. The multi-layered bandage is repeated daily during complex decongestive therapy. Since the therapist has no tool to measure the pressure developed by the bandage, the only way to verify the efficacy of the treatment is the tracking of the volume variations (hopefully a reduction) of the treated limb. Thirdly, the maintenance phase of the treatment consists of daily compression by a low-stretch elastic stocking or sleeve. Sometimes the compressive garments need to be tailored according to the single dimorphism. For all these reasons, it is crucially important to provide accurate measurements of the limb size, and therefore to check the mostly used methods and to propose new techniques. As complex decongestive therapy is still the only available treatment for lymphedema-ma, improving the quality of its two phases through accurate assessment of limb volumes would improve the efficacy of the therapy. An optimized and tailored therapy would ultimately increase the quality of life of patients in terms of esthetics, increased mobility and lower infection incidence.”  

We have also included the studies found in the literature in the discussion as follows:

“… The intraclass correlation coefficients found were generally strong, ranging from 0.91 to 1[13], [16], [20], [24], [26]. …. ”

“… We exclude temperature as a potential factor of limb volume variability as arm volumes determined at 38°C and 16°C were shown to be almost equal[23]….”

“This is an important result considering that in 2001 Megens et al concluded that water displacement volumetry was the only method to provide an accurate estimate of the volume of the upper extremity in women after axillary node dissection for breast cancer[13].”

“… This conclusion was supported by Taylor and colleagues who concluded that volumes calculated from anatomic landmarks are more accurate than those obtained from circumferential measurements based on distance from fingertips[27].”

There are some writing and grammar mistakes and they should be corrected and the English writing of the paper should be improved.

We hope to have improved the English writing. 

Reviewer 2 Report

Comments and Suggestions for Authors

The article “Limb volume measurements: a comparison of circumferential techniques and optoelectronic systems against water displacement” compares different techniques for measuring limb volume, focusing on assessing lymphedema. The study evaluates the reliability and validity of various methods against water volumetry, considered the gold standard for limb volume measurements. The techniques examined include the fixed-height and segmental centimetric methods, opto-electronic plethysmography, and the IGOODI gate technology. The results indicate that the centimetric methods and the IGOODI system are accurate in measuring limb volume, with errors below 2%. Each method has its advantages and disadvantages, and it is crucial to evaluate new objective and reliable techniques to improve diagnostic and follow-up possibilities for lymphedema patients.

Based on the content provided, here are some suggestions and comments to improve the quality of the manuscript for publishing:

  1. The study population is relatively small, and it would be beneficial to include a larger and more diverse sample to enhance the generalizability of the findings. Additionally, considering the inclusion of severe lymphedema cases could provide a more comprehensive understanding of the measurement techniques' effectiveness across a broader range of volumes and morphologies.
  2. The limitations of the study, such as the single operator for centimetric evaluations, should be acknowledged. Future studies should consider evaluating inter-operator reliability to assess the impact of different levels of experience on measurement accuracy.
  3. The comparison of different measurement methods within the same day is a strength of the study. However, it would be valuable to address the potential variability in limb volume, especially in lymphedema-affected limbs, over time. This could impact the comparison among various methods and should be considered in the interpretation of the results.
  4. The clinical implications of the study are well-addressed, particularly in the context of evaluating and monitoring edema in patients with lymphedema following cancer treatment and chronic venous insufficiency. It would be beneficial to further emphasize the potential impact of the study findings on patient care and treatment strategies.
  5. The discussion of the limitations and strengths of the optoelectronic plethysmography system provides valuable insights. However, it would be helpful to explore potential future developments or modifications that could enhance the system's accuracy and applicability for limb volume measurements.
  6. The manuscript should include a clear and detailed measurement protocol for the centimetric techniques, as well as a discussion of the potential implications of the study findings for clinical practice and patient care.

Overall, the manuscript provides valuable insights into the reliability and validity of different limb volume measurement techniques. Addressing the above points would further enhance the quality and impact of the study.

Comments on the Quality of English Language

Moderate editing of English language required

Author Response

We thank the reviewer for their interest in our manuscript and the pertinent suggestions and comments. We hope to have fulfilled all so that the quality of the manuscript improved for publishing.

Based on the content provided, here are some suggestions and comments to improve the quality of the manuscript for publishing:

  1. The study population is relatively small, and it would be beneficial to include a larger and more diverse sample to enhance the generalizability of the findings. Additionally, considering the inclusion of severe lymphedema cases could provide a more comprehensive understanding of the measurement techniques' effectiveness across a broader range of volumes and morphologies.

We have discussed this as follows: “The number of subjects was relatively small, but the dispersion of the data in the correlation plot was so small (with points squeezed around the identity line) and the resulting Pearson correlation coefficient was so close to the unity and this reinforced the results. Including lymphoedematous limbs and considering both upper and lower limbs was a strength, because we have included a wider range of volumes (ranging from 1 to 7 litres) and morphology. The absence of severe lymphoedema was a limitation, because in-creased skin folds, fat deposits, and wart-like growths, can develop. At this severe stage, the dimorphism becomes so severe (elephantiasis in the worst scenario) that it would be extremely hard to identify the points of interest for all the considered methods. In addition, patients with this kind of severe lymphedema (of the lower limb) might experience mobility difficulties that would make it challenging to complete the protocol (in particular the gold standard).”

  1. The limitations of the study, such as the single operator for centimetric evaluations, should be acknowledged. Future studies should consider evaluating inter-operator reliability to assess the impact of different levels of experience on measurement accuracy.

We have expanded this part as follows: “Considering a single operator for the centimetric evaluations was a limit, for the lack of generality of obtained effects. However, we have shown that an expert operator, following a rigorous method of measurement, can perform similarly to the gold standard and/or more complex system (like IGOODI), with an error ~ 1%. Future studies should consider also inter-operator reliability to assess the impact of different levels of experience on measurement accuracy. It would be interesting to evaluate the repeatability of the measurement but also the learning process of a “naïve” operator following a rigorous method of centimetric measurements.”

  1. The comparison of different measurement methods within the same day is a strength of the study. However, it would be valuable to address the potential variability in limb volume, especially in lymphedema-affected limbs, over time. This could impact the comparison among various methods and should be considered in the interpretation of the results.

We have addressed this as follows: “The potential variability in limb volume of lymphedema-affected limbs over time is due to many factors: 1) pathophysiology of the condition such as lymphangitis (i.e.: inflammation of the lymphatic vessels that cause lymph nodes in the groin or armpit to swollen); 2) low-quality treatment (i.e.: inappropriate bandage and/or compression garments) so that the swelling associated with the condition is not properly contained and 3) low compliance of the patient to the treatment who does not wear daily the prescribed compression garments. We exclude temperature as a potential factor of limb volume variability as arm volumes determined at 38°C and 16°C were shown to be almost equal[23].”

  1. The clinical implications of the study are well-addressed, particularly in the context of evaluating and monitoring edema in patients with lymphedema following cancer treatment and chronic venous insufficiency. It would be beneficial to further emphasize the potential impact of the study findings on patient care and treatment strategies.

We have emphasized this as follows: “The potential impact of the findings of this study on patient care and treatment strategies ranges from the diagnosis to the treatment and the follow-up of the patient. Firstly, lymphedema is classified according to the excess percentage volume differences between the affected and unaffected limbs, namely mild (5-20% increase in limb volume), moderate (20-40% increase), or severe (>40% increase)[33]. Secondly, the multi-layered bandage during the first intensive phase of the complex decongestive therapy is completely opera-tor-dependent. The multi-layered bandage is repeated daily during complex decongestive therapy. Since the therapist has no tool to measure the pressure developed by the bandage, the only way to verify the efficacy of the treatment is the tracking of the volume variations (hopefully a reduction) of the treated limb. Thirdly, the maintenance phase of the treatment consists of daily compression by a low-stretch elastic stocking or sleeve. Sometimes the compressive garments need to be tailored according to the single dimorphism. For all these reasons, it is crucially important to provide accurate measurements of the limb size, and therefore to check the mostly used methods and to propose new techniques.

As complex decongestive therapy is still the only available treatment for lymphedema-ma, improving the quality of its two phases through accurate assessment of limb volumes would improve the efficacy of the therapy. An optimized and tailored therapy would ultimately increase the quality of life of patients in terms of esthetics, increased mobility and lower infection incidence.”

  1. The discussion of the limitations and strengths of the optoelectronic plethysmography system provides valuable insights. However, it would be helpful to explore potential future developments or modifications that could enhance the system's accuracy and applicability for limb volume measurements.

We have discussed this as follows: “The first modification that could enhance the accuracy and applicability of opto-electronic plethysmography for limb volume measurements might be the use of more markers in each line. However, increasing the number of markers will reduce the distance between two markers with the risk of approaching the spatial resolution of the system that would not recognize two adjacent markers as separate objects of interest. Another important potential future development can be the use of a laser point to create active (and no more passive) markers to scan the whole limb. Of note, the use of active markers with opto-electronic plethysmography still needs to be implemented.”

  1. The manuscript should include a clear and detailed measurement protocol for the centimetric techniques, as well as a discussion of the potential implications of the study findings for clinical practice and patient care.

To make it clearer, we have now included a new Figure 3 that better supports the description and the identification of the points of interest of the segmental technique.

Overall, the manuscript provides valuable insights into the reliability and validity of different limb volume measurement techniques. Addressing the above points would further enhance the quality and impact of the study.